# Corrosion Resistance and Titanium Ion Release of Hybrid Dental Implants

**DOI:** 10.3390/ma16103650

**Published:** 2023-05-10

**Authors:** Daniel Robles, Aritza Brizuela, Manuel Fernández-Domínguez, Javier Gil

**Affiliations:** 1Department of Translational Medicine CEU, San Pablo University, Urbanización Montepríncipe, Alcorcón, 28925 Madrid, Spain; drobles@clinica.uemc.es; 2Facultad de Odontología, Universidad Europea Miguel de Cervantes, C/del Padre Julio Chevalier 2, 47012 Valladolid, Spain; 3Department of Oral and Maxillofacial Surgery, Hospital Monteprincipe, University CEU San Pablo, Av. de Montepríncipe s/n, Alcorcón, 28668 Madrid, Spain; 4Bioengineering Institute of Technology, Facultad de Medicina y Ciencias de la Salud, Universidad Internacional de Catalunya, Josep Trueta s/n, Sant Cugat del Vallés, 08195 Barcelona, Spain

**Keywords:** hybrid dental implants, titanium, ion release, corrosion, roughness, topography

## Abstract

One of the strategies for the fight against peri-implantitis is the fabrication of titanium dental implants with the part close to the neck without roughness. It is well known that roughness favors osseointegration but hinders the formation of biofilm. Implants with this type of structure are called hybrid dental implants, which sacrifice better coronal osseointegration for a smooth surface that hinders bacterial colonization. In this contribution, we have studied the corrosion resistance and the release of titanium ions to the medium of smooth (L), hybrid (H), and rough (R) dental implants. All implants were identical in design. Roughness was determined with an optical interferometer and residual stresses were determined for each surface by X-ray diffraction using the Bragg–Bentano technique. Corrosion studies were carried out with a Voltalab PGZ301 potentiostat, using Hank’s solution as an electrolyte at a temperature of 37 °C. Open-circuit potentials (Eocp), corrosion potential (Ecorr), and current density (icorr) were determined. Implant surfaces were observed by JEOL 5410 scanning electron microscopy. Finally, for each of the different dental implants, the release of ions into Hank’s solution at 37 °C at 1, 7, 14, and 30 days of immersion was determined by ICP-MS. The results, as expected, show a higher roughness of R with respect to L and compressive residual stresses of −201.2 MPa and −20.2 MPa, respectively. These differences in residual stresses create a potential difference in the H implant corresponding to Eocp of −186.4 mV higher than for the L and R of −200.9 and −192.2 mV, respectively. The corrosion potentials and current intensity are also higher for the H implants (−223 mV and 0.069 μA/mm^2^) with respect to the L (−280 mV and 0.014 μA/mm^2^ and R (−273 mV and 0.019 μA/mm^2^). Scanning electron microscopy revealed pitting in the interface zone of the H implants and no pitting in the L and R dental implants. The titanium ion release values to the medium are higher in the R implants due to their higher specific surface area compared to the H and L implants. The maximum values obtained are low, not exceeding 6 ppb in 30 days.

## 1. Introduction

For decades, the placement of dental implants to replace missing teeth has been a routine procedure in dental clinics. There is also established evidence that justifies the use of implants in partial or total rehabilitation of edentulous patients and their success in the medium to long term [1,2]. However, dental implants are not free of complications. Once osseointegrated, implants can suffer a loss of crestal marginal bone, either due to physiological bone remodeling or due to a pathological, inflammatory process associated with the presence of bacteria [3].

Peri-implantitis is a pathological condition of the tissues surrounding the implants and is characterized by inflammation of the peri-implant mucosa and the subsequent progressive loss of the bone that supports them [4]. Using this definition, recent systematic reviews have evaluated the prevalence of peri-implant disease. Derks et al., in 2015, described a prevalence of peri-implantitis of 22% (CI: 32%–54%) in 1196 patients and 4209 implants [5]. Derks et al., in 2016, performed a randomized cross-sectional analysis on the Swedish population where they reported 45% of patients with the presence of peri-implantitis after 9 years of implant loading and bone loss greater than 0.5 mm, although it decreased to 14.5 mm when bone loss greater than 2 mm was assessed [6]. Although peri-implantitis is defined as a pathological process associated with inflammation and the presence of bacteria, its etiology is complex and multifactorial [7,8,9,10].

In recent years, there have been numerous advances in surface modifications and treatment to improve osseointegration and promote BIC (Bone Implant Contact). Rough surfaces increase the survival rate of implants compared to smooth or machined surfaces [11]. However, the implant surface is also considered a potential risk factor for crestal marginal bone loss [12]. It has been demonstrated by in vitro studies that rough surfaces have more facility to develop biofilm adhered to them and that in these, it is more difficult to debride and remove it, in comparison with smooth surfaces [13,14].

In animal studies, the progression of peri-implantitis in implants with a rough surface was faster than in implants with a machined surface [15] and, clinically, there is a higher rate of peri-implantitis in implants with a rough surface than in machined surfaces [16]. It seems reasonable then, the use of moderate roughness implants (roughness with Sa = 1–2 µm values) that have a better osseointegration index and survival rate than machined surface implants, but at the same time are less prone to biofilm accumulation and subsequent bone loss caused by peri-implant disease than implants with higher roughness (Sa greater than 2 µm) [17]. However, as peri-implantitis is still significantly prevalent in these moderate roughness implants [18], hybrid implants have appeared on the market.

Hybrid implants or hybrid surface implants present a design in which the coronal area has machined titanium and the rest of the implant has a rough surface. Apparently, this design has a number of advantages over other surface designs. Both smooth and rough parts must be in contact with bone. However, more of the coronal area is made smooth so that if there is bacterial infiltration through the surface of the dental implant, it is more difficult for biofilm to form. In the case of bone loss, the removal of the biofilm by implantoplasty will be much easier as the surface is smooth. The machined coronal part reduces the presence of biofilm by reducing wettability and cell adhesion at this level, and the treated or roughened part favors osseointegration [19]. Hybrid implants have a high survival rate, marginal bone stability, and fewer biological complications [20]. However, animal studies have shown that implants with a hybrid surface present a degree of inflammation of the peri-implant tissues similar to that of implants with moderate roughness [21] and a greater marginal bone loss [22].

It is therefore interesting to determine how these implants respond to the different factors that can favor the appearance and development of peri-implant disease, and among them, to know how corrosion can affect the hybrid surface and whether or not there are differences with respect to implants with a rough surface. It is indisputable that titanium particles have been found in peri-implant tissues and play a role in the inflammatory response of the tissues and in marginal bone loss [23]. In vitro studies have shown the potential of titanium ions or particles as toxic or proinflammatory particles [24]. Inflammation can cause the medium to acidify by changing the pH conditions around the implants and producing the active dissolution of metal ions. The appearance of these ions or particles, produced by the corrosion of the material, triggers an inflammatory reaction, and this is why corrosion can be considered one of the factors associated with the appearance and establishment of peri-implant disease. The fact that the implants can be in contact with different external factors can lead to their degradation. Among these factors are electrochemical factors, acidity due to inflammation, bacteria, the use of solutions or toothpaste rich in fluoride, and mechanical factors derived from the loads [25].

The corrosion simulated in vitro with fluids and environments similar to those exposed in the oral environment has been demonstrated in different studies, and in them, the appearance of titanium ions depends on pH, the time of immersion in the fluid, and the type of acid [26]. This process is known as biocorrosion. The concept of biocorrosion can be extended if mechanical factors produced by the load or friction between the different elements of the implant–prosthesis complex are taken into account. This type of corrosion is known as tribocorrosion [27,28,29].

In summary, wear, corrosion, titanium particles, inflammation, and microorganisms are all part of a complex host response to foreign bodies. There is some relationship between corrosion, the presence of titanium particles, and biological complications. However, there is currently insufficient data to support a unidirectional role of titanium corrosion and metal particles in the pathogenesis of peri-implantitis.

The consensus document on peri-implantitis [30] explains that there is no evidence that peri-implantitis is favored by the surface of the dental implant. There are other parameters such as access to dental hygiene or geometric aspects of the abutment that may affect it. However, it is evident that cleaning treatments for bacterial biofilm are much easier on surfaces than on rough ones. On rough surfaces, implantoplasty treatments must often be performed, and on smooth surfaces, cleaning is more favored.

The aim of this contribution is to determine the corrosion resistance and ion release of hybrid dental implants. The hypothesis of the contribution is that the presence of smooth and rough parts causes an interface of different topography with an important change in residual stresses. This fact will affect the chemical degradation of the dental implant. The study of this behavior is original and can illustrate the long-term behavior of this type of implant that the clinician should take into account.

## 2. Materials and Methods

Ninety commercially pure titanium grade 3 bone-level dental implants donated by Klockner Dental Implants (Escaldes Engordany, Andorra) were studied. Batches of thirty implants were made: The first batch was machined implants, which were named Smooth (S); another thirty implants were smooth on the first three coils (4 mm) and the rest of the dental implant was shot blasted with alumina particles sized 220 μm at a pressure of 5 bars to cause the rough topography of the dental implants, and this second batch was named the hybrid implant (H); regarding the third batch, the entire surface of the dental implant was treated with alumina grit blasting under the same conditions as described above. These implants were named rough implants (R). Figure 1 illustrates the three types of dental implants. A flow chart of the study can be observed in Figure 2.

### 2.1. Roughness

White light interferometry (Wyko NT1100 Optical Interferometer, Veeco Instruments, Plainview, NY, USA), in vertical scanning interferometry mode, was used to produce, evaluate, and quantify the topography. Interferometric measurement presents a high vertical resolution (≈2 nm). The analysis area was 122.5 × 94.6 µm^2^. The results were analyzed by Wyko Vision 32 (Veeco Instruments, Plainview, NY, USA) applying the Gaussian filter to increase the resolution of the roughness. Four specimens were analyzed characterizing the amplitude parameter (Sa), the spacing parameter (Sm), and the hybrid parameter (Index area).

### 2.2. Residual Stresses

Residual stresses were determined by means of a diffractometer incorporating a Bragg–Bentano configuration (D500, Siemens, Erlangen, Germany). The measurements were studied for the planes (213). These diffract at 2θ = 139.5°. The elastic constants of titanium in the direction of (213) planes are EC = (E/1 + ν) (213) = 90.3 (1.4) GPa. Eleven Ψ angles, namely, 0° and five positive and five negative angles, were determined. The diffraction peaks were adjusted with a pseudo-Voigt function using the appropriate software (WinplotR, software version 3.2, free access online, Microsft, Redmon, WA, USA), and then converted to interplanar distances (d Ψ) using Bragg’s equation. The dΨ vs. sen2 Ψ graphs and the calculation of the slope of the linear regression (A) were performed with the appropriate software (software version 5.0, Origin, Microcal, San Mateo, CA, USA). The residual stress is: σ = EC (1/d_0_) A, where d_0_ is the interplanar distance for Ψ = 0°.

### 2.3. Corrosion Resistance

Corrosion tests were performed following the ISO and ASTM standards [31,32,33,34].

The tests were carried out with a Voltalab PGZ 301 potentiostat (Radiometer, Copenhagen, Denmark) controlled by Voltamaster 4 software (Radiometer Analytical, Villeurbanne Cedex, France). The electrolyte used in this study consisted of Hank’s solution (Sigma Aldrich, San Luis, MO, USA). The chemical composition can be observed in Table 1. This saline solution reproduces the ionic composition of the human physiological environment. The testing solution was kept at a controlled constant temperature of 37 °C.

The electrical setup used to measure the electrochemical parameters and sample geometry is represented in Figure 2. The reference electrode was an Ag/AgCl/KCl electrode (E_ = 0.222 V). The auxiliary electrode used was a platinum electrode with a surface of 240 mm^2^ (Radiometer Analytical, Villeurbanne, France).

The open circuit potential was monitored for 3 h in order to allow leveling off of the value before the polarization resistance test. The Cyclic Voltammetry assay was performed by scanning the potential of the alloy of the sample at 0.25 mV/s with the minimum current set at −1 A and the maximum at +1 A, with a minimum range set at 100 lA between −300 and +2000 mV around the OCP value. Recordings of the variation in galvanic current density, potential, etc., were obtained, and the Tafel slopes were determined from the Evans diagrams. In order to determine these diagrams, it is very important to record the polarization curves in a pseudo-stationary manner. In this case, 250 min of immersion of the specimens sufficed.

Open circuit potential (Eocp), corrosion potentials (Ecorr), and corrosion currents (icorr) were recorded for the different samples tested. These parameters are defined as:

Open circuit potential (Eocp): Potential of an electrode measured with respect to a reference electrode or another electrode when no current flows to or from the material.

Corrosion potential (Ecorr): Potential calculated at the intersection where the total oxidation rate is equal to the total reduction rate.

Corrosion current density (icorr): Current divided by the surface of the electrode. This is the size of the anodic component of the current, which flows at the corrosion potential (Ecorr).

Since, by definition, the resulting current is equal to zero at that potential, the cathodic component is of equal size but of the opposite sign. With the measured resulting current being zero at the corrosion potential, the corrosion current density icorr can only be obtained by indirect methods, e.g., from the Tafel equation. Tafel constants a and b are the Tafel proportionality constants for anodic (oxidation) and cathodic (reduction) reactions [35,36,37,38].

### 2.4. Ion Release

An ion release test was performed by immersing the implants in 100 mL of Hank’s solution at 37 °C, for 1, 7, 14, and 30 days. Ion-release quantification was carried out by inductively coupled plasma-mass spectrometry (ICP-MS) by using Perkin Elmer Optima 320RL equipment (Waltham, MA, USA). Titanium calibration standards were prepared by serial dilution containing Ti ions at different concentrations using elemental stock solutions to prepare calibration standards. Each solution extract was analyzed in triplicate and the concentrations were determined using linear regression. An ion-release test was conducted using a Memmert Incubator Oven model BE500 (MEMMERT Gmbh, Schwabach, Germany).

### 2.5. Scanning Electron Microscopy

The surfaces of the samples were observed using SEM (JEOL JSM 5410 Microscopy Tokyo, Japan) equipped with a link LZ5 EDS (Jeol, Tokyo, Japan) operated at 10 kV, which was also used for determining the chemical composition.

### 2.6. Statistical Analysis

The data were statistically analyzed using Student’s *t*-tests, one-way ANOVA tables, and Tukey’s multiple comparison tests in order to evaluate any statistically significant differences between the sample alpha = 0.05.

## 3. Results

Figure 3 shows the SEM micrographs of the different roughness values obtained. It can be seen that the smooth implant presents a lower roughness than the grit-blasting implant. The hybrid implants present a thick interface among the smooth and roughened parts of the implant.

The results of the roughness parameters obtained are shown in Table 2, and the differences obtained for roughness parameters Sa, Sm, and the index area confirmed that there were statistically significant differences (*p* < 0.05) between rough and smooth samples.

Table 3 confirms the compressive character of the residual stresses for the grit-blasting dental implants and the part of the hybrid dental implant submitted to the abrasive projection. As expected, the compressive stresses induced by grit blasting are statistically significant (*p* < 0.001, *t*-Student) and highly different from smooth dental implants.

The results obtained by performing potentiostatic tests under open-circuit conditions (OCP), in which the open-circuit potential (EOCP) was determined after 16 h of immersion without applying current, are presented below. Table 4 shows the open-circuit potential for each dental implant: Smooth (S), Hybrid (H), and Rough (R). Table 4 presents the mean values of open-circuit potential determined for each of the three groups of samples analyzed, expressing the result as a mean value and its corresponding standard deviation.

The results obtained by performing the anodic polarization potentiodynamic tests are presented, in which the corrosion potential (ECORR) and corrosion intensity (iCORR) were determined from the polarization curves. The graph in Figure 4 shows three representative potentiodynamic corrosion curves for the three types of samples evaluated.

Table 5 presents the mean corrosion potential and current density values determined for each of the three groups of samples analyzed, expressing the result as a mean value and its corresponding standard deviation.

Figure 5 graphically shows the level of titanium ions released into the electrolyte expressed in concentration [ppb] and its corresponding standard deviation. It is observed that the ones that release more ions are the rough dental implants since the specific surface of the implant is much higher than in the other two cases.

## 4. Discussion

As expected, dental implants subjected to abrasive spraying show a higher roughness than polished implants in both Sa and Sz values. From studies by other authors [39,40,41], it can be assessed that the specific surface increases compared to the smooth one by a factor of 8 to 10. This increase in potential surface area in contact with the bone tissue is what guarantees good bone anchorage and therefore biological fixation of the dental implant. However, this roughness that favors adhesion, proliferation, and osteoblastic differentiation [39,40,41,42] facilitates bacterial colonization [9,43,44,45,46,47,48], and it is for this reason that new hybrid implants have been developed. In consequence, the implants have a polished part on the surface to hinder the adhesion of bacteria and the formation of biofilm. In other words, the purpose of hybrid implants is to avoid peri-implantitis by sacrificing osseointegration.

As it has been proven, having a rough surface generates a compressive residual stress of −201.2 MPa, and this residual stress improves the mechanical properties of the dental implant. These values are very similar to those obtained by other authors on shot-blasted titanium implants [49,50]. These residual stress values depend on the projection pressure, the distance from the gun to the surface, and the nature of the abrasive material. In general, these values range from −150 to −270 MPa, being compressive in all cases. Having a compressive surface hinders the appearance of fatigue cracks on the surface of the dental implant and has to do so at a depth of approximately 10 μm, delaying the onset of fatigue cracks, and therefore the fatigue life of the dental implant exceeds the expected life of the patient [50,51]. It is well known that the same material, where one part is under residual stress and the other part is not, generates a corrosion potential at the interface. The area most sensitive to pitting will be the stressed area and especially the smooth–rough area, which is where the material has the greatest corrosion potential. These pits can be fatigue crack initiation zones [52]. In our case, the hybrid dental implants have the rough part subjected to a compressive stress of −201.2 MPa and the smooth part to −20.2 MPa, and this mechanical heterogeneity causes a corrosion potential. This fact has been demonstrated with open-circuit potentials where hybrid dental implants show higher values than smooth or rough ones [53,54]. These differences between them are statistically significant. The dental implants with the highest resistance to corrosion are the smooth ones, followed by the rough ones, and, finally, the hybrid ones due to the heterogeneity of residual stresses in the body of the dental implant.

The results of the potentiostatic curves show, as expected, higher values of current intensity and corrosion potentials for the hybrid dental implants. As can be seen in Figure 6, numerous pitting instances can be observed in the interface zone of the hybrid dental implant between the smooth and rough parts, since this is the zone with the highest localized corrosion potential. Figure 7 shows scanning electron microscopy images of the smooth and rough surfaces of areas of the hybrid implant away from the interface, where no pitting can be seen. These results confirm the hypotheses of this contribution.

These pits, in addition to the possible toxicity of the corrosion products, can be fatigue crack initiation zones [52]. Smooth and rough dental implants subjected to corrosion did not have observable pitting areas on the surface of the dental implant.

The results of ion release show higher values at all tested times, indicating that rough dental implants have a larger specific surface area than hybrid dental implants and even larger than smooth dental implants. For this reason, the values of titanium ions released into the medium are higher. However, it should be noted that the values are very small, parts per billion, and therefore there does not seem to be any toxicity problem. It is important to distinguish chemical degradation by corrosion, which is a chemical reaction, from degradation by ion release, which is a solution of metal ions to the medium [55,56].

The philosophy of hybrid dental implants is controversial from biological and microbiological perspectives. These implants have a smooth coronal area to facilitate bacterial decontamination processes and avoid implantoplasty in the case of biofilm formation [53]. However, the loss of roughness causes there to be a less specific surface area for bone tissue formation, and therefore the mechanical fixation of hard tissue will be less. In this research work, we have been able to verify how the corrosion values in both open and potentiodynamic potentials are worse than rough and smooth ones. It was possible to observe pitting at the interface, which is a suitable place for the initiation of a fatigue crack. The difference in residual stresses between both surfaces causes this behavior of chemical degradation that can be favored if there are other metals or alloys in the oral cavity [40,41]. We have been able to verify that the corrosive effect does not significantly affect the release of ions into the physiological environment, with low concentration values. However, the corrosion products generated should be studied, especially at the rough–smooth surface interface, which is supposed to be mixed oxides of titanium, and their potential cytocompatibility should be studied.

These results make us reflect on the care that the clinician must take into account in the placement of hybrid dental implants. Bacterial hindrance to prevent biofilm formation sacrifices the good osseointegration capacity of rough surfaces and, as we have seen, will affect the long-term electrochemical corrosion behavior of the implant [57]. Different disinfection methods have been studied for titanium surfaces to improve their bactericidal behavior. Research by Alovisi et al. [58] showed the effect of glycine powder sprayed with a mixture of antibiotics, with very encouraging results. Further research is needed to achieve a dental implant with a bacteriostatic and/or bactericidal surface without affecting the corrosion of the dental implant.

## 5. Conclusions

The values of open-corrosion potential, current density, and corrosion potential show that hybrid implants have the worst values, offering less resistance to corrosion. In general, when the implant has more surface in contact with the electrolyte, it has greater corrosion, and therefore rough dental implants should be the ones with the most corrosion. However, the difference in residual stress on the surface of −201.2 to −20.2 causes worse corrosion behavior. This difference in stress results in a decrease in the corrosion resistance of the dental implant creating pitting at the interface. The release of titanium ions is higher in the rough zone of the implant, but the values are low in the order of parts per billion. These results should be considered by clinicians in the choice of dental implants.

## Figures and Tables

**Figure 1 materials-16-03650-f001:**
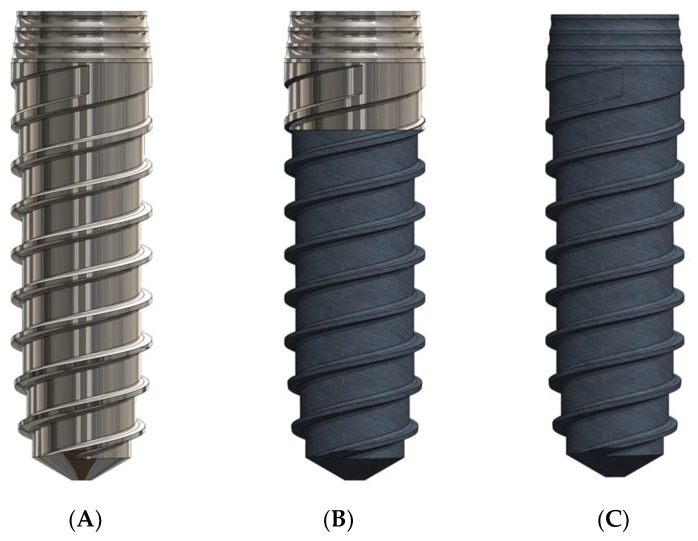
(**A**). Smooth dental implant (L). (**B**). Hybrid dental implant (H). (**C**). Rough dental implant (R).

**Figure 2 materials-16-03650-f002:**
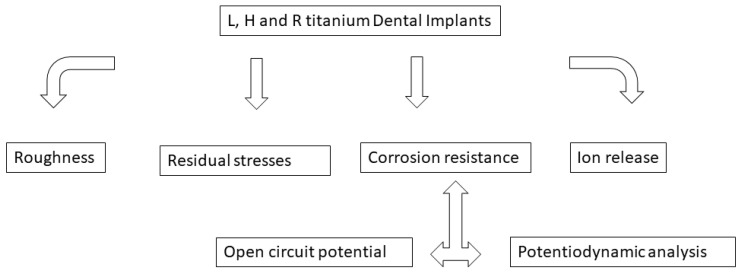
Flow chart of the study.

**Figure 3 materials-16-03650-f003:**
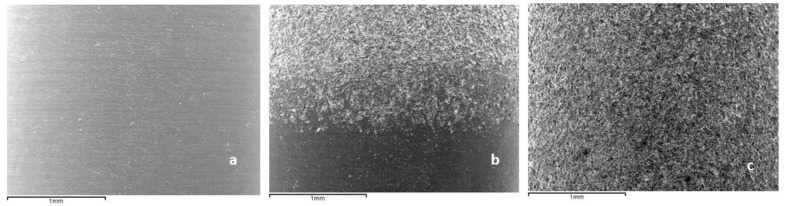
Surfaces of different dental implants studied. (**a**). Smooth dental implant. (**b**). Hybrid implant. (**c**). Roughened implant.

**Figure 4 materials-16-03650-f004:**
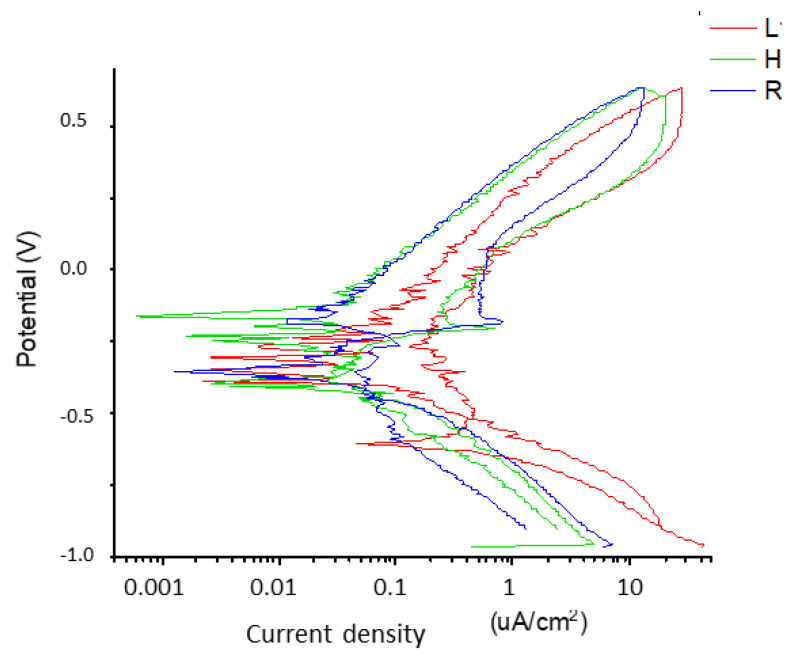
Potentiodynamic curves of different dental implants studied.

**Figure 5 materials-16-03650-f005:**
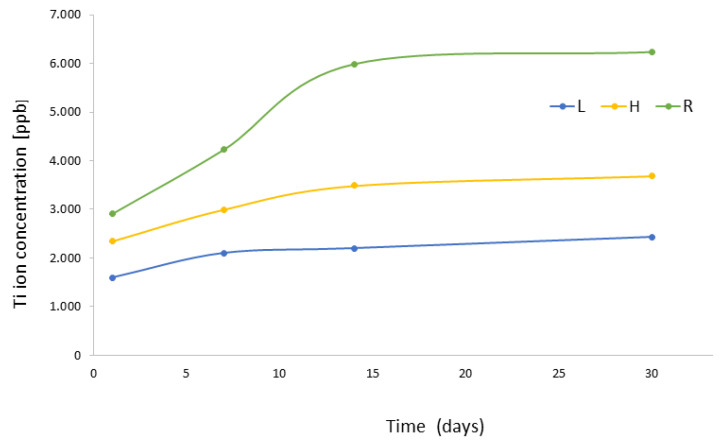
Titanium ion release of the different dental implants studied in Hank’s solution after 1, 7, 14, and 30 days of immersion.

**Figure 6 materials-16-03650-f006:**
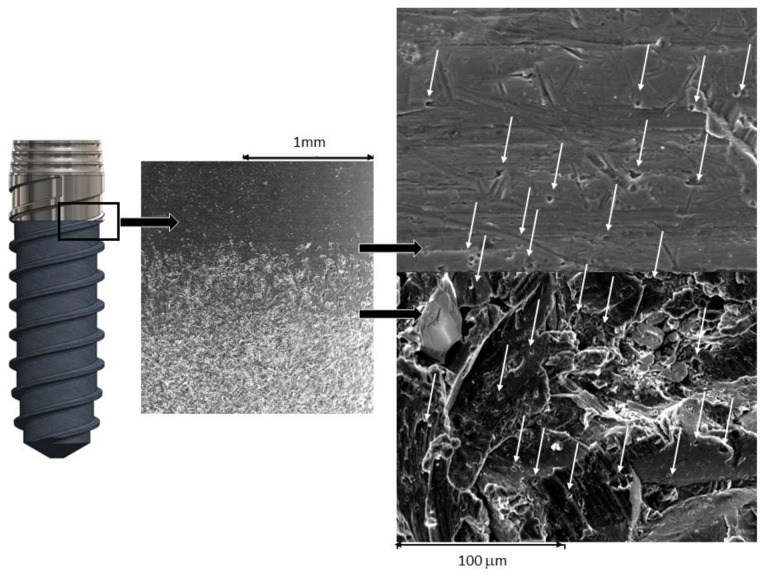
Scanning electron microscopy observation in the interface (smooth–roughness) where pitting (white arrows) on the titanium surfaces can be observed.

**Figure 7 materials-16-03650-f007:**
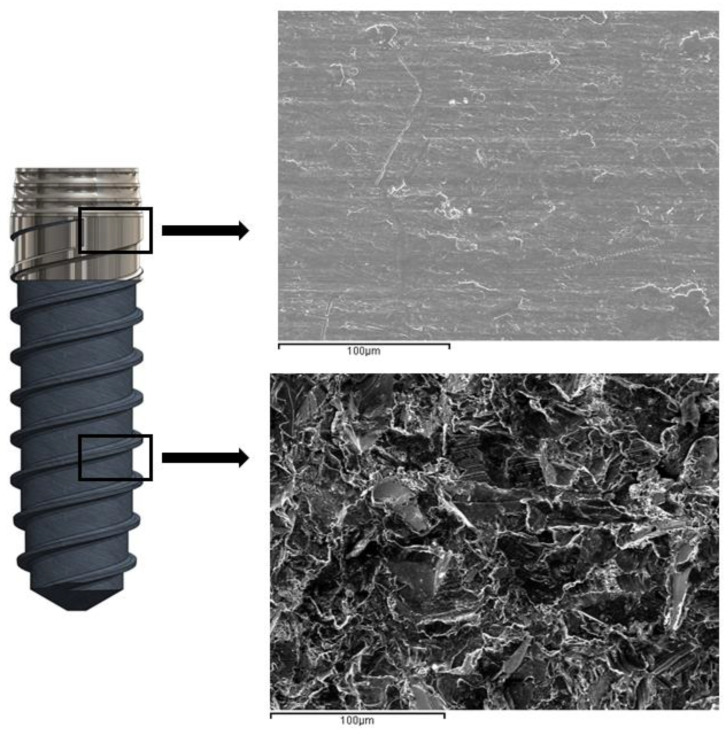
Scanning electron microscopy observation in the smooth zone and the roughness zone far from the interface. Pitting is not observed.

**Table 1 materials-16-03650-t001:** Chemical composition of Hank’s solution.

Component	Composition (mM)
K_2_HPO_4_	0.44
KCl	5.4
CaCl_2_	1.3
Na_2_HPO_4_	0.25
NaCl	137
NaHCO_3_	4.2
MgSO_4_	1.0
C_6_H_12_O_6_	5.5

**Table 2 materials-16-03650-t002:** Roughness and area index values for smooth and rough surfaces. Asterisk indicates statistically significant differences between the surfaces studied by ANOVA in each roughness parameter (*p* < 0.05).

Surface	Sa (µm) ± SD	Sm (µm) ± SD	Index Area ± SD
Smooth	0.23 ± 0.02	0.33 ± 0.01	1.10 ± 0.02
Rough	1.98 ± 0.12 *	5.40 ± 0.20 *	1.16 ± 0.05 *

**Table 3 materials-16-03650-t003:** Surface residual stresses calculated at the smooth and rough surfaces of the dental implant (* means statistically significant difference).

Dental Implant	Residual Stress (MPa)
Smooth	−20.2 (5.3)
Rough	−201.2 (11.2) *

**Table 4 materials-16-03650-t004:** Open circuit potential for each type of dental implant. Asterisk and double asterisk indicate statistically significant differences between the three surfaces studied in each roughness parameter (*p* < 0.05) obtained by ANOVA.

Dental Implants	E_OCP_ (mV)
L	−200.9 ± 13.3
R	−192.2 ± 0.10 *
H	−186.4 ± 4.9 **

**Table 5 materials-16-03650-t005:** Values of current density and corrosion potential of different dental implants studied. Symbols indicate statistically significant differences between the three surfaces studied in each roughness parameter (*p* < 0.05) obtained by ANOVA.

Dental Implants	j_CORR_ (µA/cm^2^)	E_CORR_ (mV)
L	0.014 ± 0.055	−280 ± 53
R	0.019 ± 0.019	−273 ± 34
H	0.069 ± 0.015 *	−223 ± 50 *

## Data Availability

The data that support the findings of this study are available from the corresponding author upon reasonable request.

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
