# Peer review of "Corrosion Resistance and Titanium Ion Release of Hybrid Dental Implants"

_materials, 2023, doi:10.3390/ma16103650_

Round 1

Reviewer 1 Report

The manuscript describes the study of the corrosion resistance and the release of titanium ions to the medium of smooth (L), hybrid (H) and rough (R) dental implants.

The scope is narrow and the methods are appropriate to achieve the objectives. Novelty and significance statements are missing.

The missing operating parameters reduces reproducibility.

The manuscript is written for a specialized dentistry journal. To qualify publication in Materials, a general materials science journal, the scope needs to be widened. For example, rather than "observe and report," please discuss the underlying mechanism.

Thus a major revision is recommended before further consideration.

Some suggestions:

1. Fig. 1 caption is not in English.

2. English usage needs to be revised, e.g. L177, L193, L316-317, L336-337, rewrite.

3. Fig. 2 content is not a result. Delete.

4. ICP operating parameters are missing.

5. Section 2.6, is it 0.005? What is alfa value?

6. Please double check all Tables. Table 2 presents only 2 samples. Table 3, title writes “four” types and presents on 2. Some use symbols in column Dental implants, and some don’t.

7. Figures 4 and 5 should use the same color theme.

8. How does -201,2 (±11,2) MPa compare to the literature? It is not -200 MPa as was written in L312.

9. “These pits…because they can be fatigue crack initiation zones.” needs a reference. This reviewer suggests DOI: 10.3390/met12030406

10. What does “peli-greasy” mean?

11. “Scanning electron microcopy has been able to observe…” Does SEM observe or do the authors observe through SEM?

12. Do what are your overall recommendations?

English usage needs to be revised, e.g. L177, L193, L316-317, L336-337, rewrite.

Fig. 1 caption is not in English.

Author Response

REVIEWER 1

Dear Reviewer,

Thanks for taking the time to review our manuscript and suggest to us to improve our work by providing a lot more detail. We have done so, and we are now submitting a manuscript that not only addresses the points the you specifically raised but also many others that we have considered in order to deliver what we think is a much improved version of our work. This version includes more paragraphs, English grammar revisions in all main sections, new references. Thanks a lot. We are looking forward to your comments.

Sincerely,

Francisco-Javier Gil Mur

The manuscript describes the study of the corrosion resistance and the release of titanium ions to the medium of smooth (L), hybrid (H) and rough (R) dental implants.

The scope is narrow and the methods are appropriate to achieve the objectives. Novelty and significance statements are missing. The missing operating parameters reduces reproducibility.

The manuscript is written for a specialized dentistry journal. To qualify publication in Materials, a general materials science journal, the scope needs to be widened. For example, rather than "observe and report," please discuss the underlying mechanism.

Thank you very much for you suggestion. In the introduction, the novelty and importance of the research work as well as the hypotheses of the work have been introduced more clearly. The new appearance of this type of dental implants in clinics has been realised only by biological and microbiological studies but there is a lack of study on their corrosion behaviour and the release of metal ions.

Thus a major revision is recommended before further consideration.

Some suggestions:

  1. 1 caption is not in English.

Done

  1. English usage needs to be revised, e.g. L177, L193, L316-317, L336-337, rewrite.

Revised

  1. 2 content is not a result. Delete.

Figure has been deleted.

  1. ICP operating parameters are missing.

More details have been introduced in the text (Materials and methods)

  1. Section 2.6, is it 0.005? What is alfa value?

Yes, this aspect has been clarified in the text

  1. Please double check all Tables. Table 2 presents only 2 samples. Table 3, title writes “four” types and presents on 2. Some use symbols in column Dental implants, and some don’t.

Roughness and residual stress tables can only be made from smooth and rough surfaces as the interface is a line. However, corrosion results can be made from the smooth zone, the rough zone and the interface zone between the smooth and the rough part. This is the reason for the different rows in the tables. The symbols refer to statistically significant differences with a p<0.005 as indicated in the text and as indicated in the guidelines of the journal.

  1. Figures 4 and 5 should use the same color theme.

The authors have tried to correct the colours but the programme does not allow us to change the colours as it does it automatically. We have clarified better in the text.

  1. How does -201,2 (±11,2) MPa compare to the literature? It is not -200 MPa as was written in L312.

The text has been revised

Corrected

  1. “These pits…because they can be fatigue crack initiation zones.” needs a reference. This reviewer suggests DOI: 10.3390/met12030406

The reference has been introduced. Thank you for your suggestion.

  1. What does “peli-greasy” mean?

Corrected

  1. “Scanning electron microcopy has been able to observe…” Does SEM observe or do the authors observe through SEM?

Corrected

  1. Do what are your overall recommendations?

The recommendations have been introduced in the text.

Comments on the Quality of English Language

English usage needs to be revised, e.g. L177, L193, L316-317, L336-337, rewrite.

Revised

Fig. 1 caption is not in English.

 Revised

Reviewer 2 Report

review attached

Need to improve the quality of English and correct so much grammar mistakes!

Author Response

REVIEWER 2

Dear Reviewer,

Thanks for taking the time to review our manuscript and suggest to us to improve our work by providing a lot more detail. We have done so, and we are now submitting a manuscript that not only addresses the points you specifically raised but also many others that we have considered in order to deliver what we think is a much improved version of our work. This version includes more paragraphs, English grammar revisions in all main sections, new references. Thanks a lot. We are looking forward to your comments.

Sincerely,

Francisco-Javier Gil Mur

In this manuscript, “Corrosion resistance and titanium ion release of hybrid dental implants”, the authors discussed the corrosion resistance and the release of titanium ions to the hybrid implants. The authors found pitting zones at the interface between the smooth and rough areas of hybrid implants and higher proportion of titanium ion release in rough dental implants. The idea of manuscript is interesting; however, it seems to clarify many problems before the decision in the Journal of Materials.

Q1. There existed many grammar mistakes, for example, on page 5 lines 194 “the Hnak’s solution”, and on page 9 lines 302 “other authors xxx”, on page 7 lines 259 “Roughned”. Please correct all these mistakes and make the paper more readable.

These grammar mistakes have been revised. Thank you very much.

Q2. On Figure 1, the authors named three different implants: S, H and R. However, on Table 4, Figure 4 and 5, S group may be changed to L group.

Corrected

Q3. In the manuscript, the authors chose Student’s t-tests, one-way ANOVA and Turkey’s multiple comparison tests. However, the authors DO NOT identify which kind of statistical analysis in each table or figure. Moreover, it is strange that each group has statistical significance on Table 4. Please clarify it.

Done

Q4. The hybrid implants are universally used in dental clinic. However, the smooth part is initially designed for attached gingiva, not for bone. Besides, the manuscript only demonstrated titanium ions release in Hank’s solution and corrosion behavior. However, the relationship between corrosion, titanium ions and potential biological complications cannot be clearly demonstrated.

Hybrid dental implants have a smooth part but it is not meant to be in the gingival part but should also be in contact with the bone. This smooth area is made so that in case of bacterial filtration it is more difficult for them to adhere.  In the case of peri-implantitis, there will be a loss of bone in the smooth area and it will be easier to disinfect it, thus avoiding implantoplasty. This aspect has been better explained in the text to avoid confusion.

Q5. As we all know, the hybrid implants have rough surface and smooth surface, which are designed for better osseointegration and anti-bacterial adhesion separately. So why the authors try to explain the corrosion of rough surface since the rough surface should be contacted with bone. Besides, why the authors compare two different environments via using Hank’s solution, since the smooth surface is attached with gingiva and the rough one contacted with bone?

The reviewer is right. Corrosion is a chemical reaction, i.e. there are reactants and a corrosion product and the release of ions is a dissolution. We have discussed this aspect in the text. As the two parts, rough and smooth, will be in contact with the bone tissue, we have carried out the corrosion tests as most authors do, which is the Hank solution and as indicated in the standard.

Reviewer 3 Report

The authors aimed to determine the corrosion resistance and ion release of hybrid dental implants. In fact, the presence of a smooth and a rough part causes an interface of different topography but more importantly of a change of residual stresses which can affect the chemical degradation of the dental implant.

The study covers some issues that have been overlooked in other similar topics. The structure of the manuscript appears adequate and well divided in the sections. Moreover, the study is easy to follow, but some issues should be improved. Some of the comments that would improve the overall quality of the study are:

I-) Authors must pay attention to the technical terms acronyms they used in the text

II-) Some references are too dated. This reviewer suggest to replace with more updated (i.e., please see : doi: 10.3390/jfb14010014).

II-) Conclusion Section: This paragraph required a general revision to eliminate redundant sentences and to add some "take-home message".

Minor editing of English language required.

Author Response

REVIEWER 3

Dear Reviewer,

Thanks for taking the time to review our manuscript and suggest to us to improve our work by providing a lot more detail. We have done so, and we are now submitting a manuscript that not only addresses the points the you specifically raised but also many others that we have considered in order to deliver what we think is a much improved version of our work. This version includes more paragraphs, English grammar revisions in all main sections, new references. Thanks a lot. We are looking forward to your comments.

Sincerely,

Francisco-Javier Gil Mur

The authors aimed to determine the corrosion resistance and ion release of hybrid dental implants. In fact, the presence of a smooth and a rough part causes an interface of different topography but more importantly of a change of residual stresses which can affect the chemical degradation of the dental implant.

The study covers some issues that have been overlooked in other similar topics. The structure of the manuscript appears adequate and well divided in the sections. Moreover, the study is easy to follow, but some issues should be improved. Some of the comments that would improve the overall quality of the study are:

I-) Authors must pay attention to the technical terms acronyms they used in the text

The authors have revised all acronyms of the text.

II-) Some references are too dated. This reviewer suggest to replace with more updated (i.e., please see : doi: 10.3390/jfb14010014).

The authors have introduced new references. The suggested by the reviewer has been also introduced.

II-) Conclusion Section: This paragraph required a general revision to eliminate redundant sentences and to add some "take-home message".

Conclusion section has been revised according to the reviewer. In this new version, the conclusion are clear and concrete. Thank you for your comment.

Reviewer 4 Report

Introduction:

- The state of the art is present, but the introduction is a little bit too long. Try to shorten it and be more concise.

- Add the study hypotheses after the aim of the study

Materials and methods

- Figure 1, the caption of the figure is currently written in a different language. Please correct it and write it in English.

- I suggest you to add the a flow chart of the study at the beginning of the M&M section to give the readers an immediately understanding of what procedures and how the study was designed. 

Discussion

- Discuss if the study hypotheses were accepted or rejected based on the results of the study

- Improve the discussion by discussing the results of other Authors who assessed the effect of different titanium surfaces in the treatment of periimplantitis and bacteria adhesion. For this porpose, discuss and cite the following recently published article

Alovisi, M.; Carossa, M.; Mandras, N.; Roana, J.; Costalonga, M.; Cavallo, L.; Pira, E.; Putzu, M.G.; Bosio, D.; Roato, I.; Mussano, F.; Scotti, N. Disinfection and Biocompatibility of Titanium Surfaces Treated with Glycine Powder Airflow and Triple Antibiotic Mixture: An In Vitro Study. Materials 202215, 4850. https://doi.org/10.3390/ma15144850

Author Response

REVIEWER 4

Dear Reviewer,

Thanks for taking the time to review our manuscript and suggest to us to improve our work by providing a lot more detail. We have done so, and we are now submitting a manuscript that not only addresses the points the you specifically raised but also many others that we have considered in order to deliver what we think is a much improved version of our work. This version includes more paragraphs, English grammar revisions in all main sections, new references. Thanks a lot. We are looking forward to your comments.

Sincerely,

Francisco-Javier Gil Mur

Introduction:

- The state of the art is present, but the introduction is a little bit too long. Try to shorten it and be more concise.

The introduction has been reduced.

- Add the study hypotheses after the aim of the study

Done

Materials and methods

- Figure 1, the caption of the figure is currently written in a different language. Please correct it and write it in English.

Done

- I suggest you to add the a flow chart of the study at the beginning of the M&M section to give the readers an immediately understanding of what procedures and how the study was designed. 

The flow chart has been introduced as Figure 2

Discussion

- Discuss if the study hypotheses were accepted or rejected based on the results of the study.

Done

- Improve the discussion by discussing the results of other Authors who assessed the effect of different titanium surfaces in the treatment of periimplantitis and bacteria adhesion. For this porpose, discuss and cite the following recently published article

Alovisi, M.; Carossa, M.; Mandras, N.; Roana, J.; Costalonga, M.; Cavallo, L.; Pira, E.; Putzu, M.G.; Bosio, D.; Roato, I.; Mussano, F.; Scotti, N. Disinfection and Biocompatibility of Titanium Surfaces Treated with Glycine Powder Airflow and Triple Antibiotic Mixture: An In Vitro Study. Materials 202215, 4850. https://doi.org/10.3390/ma15144850

The reference has been introduced and the treatment against periimplantitis discusses according to the reviewer

Round 2

Reviewer 1 Report

The revision has addressed the majority of the concerns.

some minor points, spelling "diffractometer".

2.6, alfa should be 0.05, seeing the statistics used in the later sections. Please correct it to alfa=0.05.

Table 2 title is problematic. It is not clear which part of which inplant is "Smooth", or "Rough", thus it is hard to relate the results of Table 2 and Table3 to the overall study. The "Found specimens" have not been clearly stated in the text. Are they L, H, R?

Maybe the math was a proble, how can 60 implants be made into 3 batches of 30 implants each?

English usage: "the third batch of thirty dental implants were subjected their whole body with the shot blasting treatment..."

it needs to be further improved.

Author Response

REVIEWER 1

Thank you very much for your comment and suggestions. These have been corrected and improved the text. Thank you again for your help and patience.

The revision has addressed the majority of the concerns.

some minor points, spelling "diffractometer".

Corrected

2.6, alfa should be 0.05, seeing the statistics used in the later sections. Please correct it to alfa=0.05.

Corrected

Table 2 title is problematic. It is not clear which part of which inplant is "Smooth", or "Rough", thus it is hard to relate the results of Table 2 and Table3 to the overall study. The "Found specimens" have not been clearly stated in the text. Are they L, H, R?

The reviewer is right that there were errors in the legend and could confuse readers. The legends in table 2 and 3 of the manuscript have been corrected.

Maybe the math was a proble, how can 60 implants be made into 3 batches of 30 implants each?

It is not a problem of the mathematics but of the authors. We made a mistake with the two surfaces 30+30 but there are three types of implants and they are 90. We apologize for the error which has been corrected in the text.

English usage: "the third batch of thirty dental implants were subjected their whole body with the shot blasting treatment..."

Corrected

Reviewer 2 Report

Firstly, a hybrid implant is not the most common choice. During the consensus report (https://doi.org/10.1111/clr.13827), the experts raised a clinical recommendation: “According to the present body of literature, there is no implant surface or material which has been shown to reduce the risk for peri-implantitis.”So the design of this manuscript is not so convincing. 

Secondly, the statistical methods need to be carefully correct. A mistake like “p<0.005” cannot be neglected, for the authors still do not explain the meaning behind “*,**,***” in the methods or legends. 

Thirdly, this research aims to give clinicians a hint about hybrid dental implants. However, the manuscript lacked the most critical experiments proving that the released ions do not harm cells or that the bacteria prefer the rough surface to the smooth surface.

none

Author Response

REVIEWER 2

Thank you very much for your suggestions and comments.

Firstly, a hybrid implant is not the most common choice. During the consensus report (https://doi.org/10.1111/clr.13827), the experts raised a clinical recommendation: “According to the present body of literature, there is no implant surface or material which has been shown to reduce the risk for peri-implantitis.”So the design of this manuscript is not so convincing.

Thank you very much for your comment and to help the understanding and objective of the article the authors have added a paragraph to better explain this fact.

When reading the consensus paper we can indeed extract that among all the studies no definitive conclusion can be drawn about the influence of peri-implantitis on rough or smooth surfaces. Other parameters are more important such as dental hygiene, abutment geometry, ...

Limited clinical data did not show differences between modified and non-modified implant surfaces in incidence or progression of peri-implantitis (SR). There is some evidence that restricted accessibility for oral hygiene and an emergence angle of >30 combined with a convex emergence profile of the abutment/prosthesis are associated with an increased risk for peri-implantitis (CR). Reconstructive therapy for peri-implantitis resulted in significantly less soft-tissue recession, when compared with access flap. Implantoplasty or the adjunctive use of a barrier membrane had no influence on the extent of peri-implant mucosal recession following peri-implantitis treatment (SR).

However, ease of biofilm removal as well as hygiene is always preferable on smooth surfaces.  In other words, the consensus speaks of the incidence but not of the ease of treatment and this is the reason for the birth of hybrid implants. They are thought for designs in which implantoplasty would not be necessary. I have consulted this fact with one of the authors of the consensous and he confirmed that this was the spirit of the consensus.

These hybrid implants are appearing in the implant market, such as the Ticare implants https://www.ticareimplants.com/. Personally, I believe that it is better to have dental implants that prevent bacterial adhesion rather than having a smooth surface on the coronal part to avoid an implantoplasty assuming that peri-implantitis will occur.

The authors believe that it is advisable to study other aspects of long-term behavior, such as corrosion, in which a worse behavior is observed with respect to other dental implants. 

The authors have added a paragraph in the introduction with the bibliographic citation of the consensus along the lines commented by the reviewer. Thank you very much for your contribution

Secondly, the statistical methods need to be carefully correct. A mistake like “p<0.005” cannot be neglected, for the authors still do not explain the meaning behind “*,**,***” in the methods or legends.

The text has been changed according to the reviewer 1 and the text has been changed in all Table legends.

Thirdly, this research aims to give clinicians a hint about hybrid dental implants. However, the manuscript lacked the most critical experiments proving that the released ions do not harm cells or that the bacteria prefer the rough surface to the smooth surface.

Some literature citations have been added that address that the levels of ions are too small for there to be any indication of toxicity. Toxicity due to ion release is very complicated, as there are only studies in the literature of the effects on animals but not comparable to humans. Regarding cellular and behavioral assays of bone tissue, we are preparing another contribution in which we can demonstrate that the rough surface shows a higher bone index contact at 4 and 10 weeks. These are results that have already been pointed out by different authors, not only because of the cellular behavior but also because the dental implant has a greater specific surface susceptible to be in contact with bone. It is for this reason that the great majority of dental implants have a rough surface.

Many thanks to the reviewer for his suggestion.

Reviewer 4 Report

The authors reviewed the manuscript correctly.

Author Response

Thank you very much for your help and your consideration.